# 3D imaging of undissected optically cleared *Anopheles stephensi* mosquitoes and midguts infected with *Plasmodium* parasites

**Mariana De Niz**[1¤a]*, **Jessica Kehrer**[2], **Nicolas M. B. Brancucci**[3¤b], **Federica Moalli**[4¤c], **Emmanuel G. Reynaud**[5], **Jens V. Stein**[6], **Friedrich Frischknecht**[2]

**1** Institute of Cell Biology, Heussler Research Group, University of Bern, Bern, Switzerland, **2** Center for Infectious Diseases, Integrative Parasitology, Heidelberg University Medical School, Heidelberg, Germany, **3** Wellcome Centre for Integrative Parasitology, Institute of Infection, Immunity & Inflammation, University of Glasgow, Glasgow, United Kingdom, **4** Theodor Kocher Institute, University of Bern, Bern, Switzerland, **5** School of Biomolecular and Biomedical Science, University College Dublin, Dublin, Ireland, **6** Department of Oncology, Microbiology and Immunology, University of Fribourg, Fribourg, Switzerland

¤a Current address: Instituto de Medicina Molecular João Lobo Antunes, Faculty of Medicine, University of Lisbon, Lisbon, Portugal
¤b Current address: Swiss Tropical and Public Health Institute, University of Basel, Basel, Switzerland
¤c Current address: San Raffaele Scientific Institute, Milan, Italy
* mariana.deniz@medicina.ulisboa.pt

**Data Availability Statement:** All relevant data are within the manuscript and its Supporting Information files.

## Abstract

Malaria is a life-threatening disease, caused by Apicomplexan parasites of the *Plasmodium* genus. The *Anopheles* mosquito is necessary for the sexual replication of these parasites and for their transmission to vertebrate hosts, including humans. Imaging of the parasite within the insect vector has been attempted using multiple microscopy methods, most of which are hampered by the presence of the light scattering opaque cuticle of the mosquito. So far, most imaging of the *Plasmodium* mosquito stages depended on either sectioning or surgical dissection of important anatomical sites, such as the midgut and the salivary glands. Optical projection tomography (OPT) and light sheet fluorescence microscopy (LSFM) enable imaging fields of view in the centimeter scale whilst providing micrometer resolution. In this paper, we compare different optical clearing protocols and present reconstructions of the whole body of *Plasmodium*-infected, optically cleared *Anopheles stephensi* mosquitoes and their midguts. The 3D-reconstructions from OPT imaging show detailed features of the mosquito anatomy and enable overall localization of parasites in midguts. Additionally, LSFM imaging of mosquito midguts shows detailed distribution of oocysts in extracted midguts. This work was submitted as a pre-print to *bioRxiv*, available at https://www.biorxiv.org/content/10.1101/682054v2.

## Introduction

Arthropod-borne diseases constitute an enormous public health burden world-wide. Some of the most medically relevant diseases in tropical areas caused by mosquitoes include malaria,

**Funding:** This work was supported by the European Union's Seventh Framework Programme (FP7/2007-2013) under grant agreement 242095: EVIMalar (to VH and MDN; ec.europa.eu/growth/sectors/space/research/fp7_en), by a Swiss National Foundation Grant (310030_159519 to VH; http://www.snf.ch/en/funding/Pages/default.aspx), by an ERC Starting Grant (StG 281719 to FF; www.erc.europa.eu/funding/starting-grants); and by the Chica and Heinz Schaller Foundation (to FF; www.chs-stiftung.org). The funders had no role in study design, data collection and analysis, decision to publish, or preparation of the manuscript.

**Competing interests:** The authors have declared that no competing interests exist.

dengue, yellow fever, Chikungunya fever, Zika fever, encephalitis, and filariasis [1–4]. The blood-sucking behavior of female mosquitoes is necessary for egg development and constitutes the link to vertebrate hosts, as pathogens are transmitted during mosquito blood meals. There are approximately 3,500 species of mosquitoes grouped into two main sub-families and 41 genera [5]. The two subfamilies are the *Anophelinae* and the *Culicinae*, which not only display important anatomical and physiological differences, but vary in their clinical significance as disease vectors of the pathogens they transmit. Recent outbreaks of Zika and dengue fever, as well as the constant pressure of malaria on many regions of the developing world continue to demand a better understanding of host-pathogen interactions in the vector. Advances in this field are likely to inform researchers across various disciplines about improved ways of blocking pathogen transmission. In this paper we explore 3D imaging of intact (in contrast to dissected), optically cleared *Anopheles* mosquitoes as vectors for the *Plasmodium* parasite, the causing agent of malaria. We envisage that the technique is equally useful to *Aedes* and *Culex* mosquitoes, both of which are important vectors of a wide range of pathogens.

Malaria causes over 200 million infections and over 400,000 human deaths per year [6]. Although hundreds of vertebrate-infecting *Plasmodium* species exist, only five species are infectious to humans. During their life cycle, *Plasmodium* parasites adopt various forms, both invasive and replicative, within the vertebrate host and the mosquito vector (reviewed by [7,8]). While rodent-infecting parasites have been imaged in all relevant tissues within mice (skin, liver, blood and bone marrow) [9–12], imaging of parasites within the living mosquito has remained largely elusive and limited to the passive floating of sporozoites in the hemolymph and proboscis [13,14]. The development of sporozoites *in vivo* in the midgut and their entry into mosquito salivary glands remains to be visualized. As an optically opaque cuticle surrounds these organs, most of the imaging achieved so far has relied on dissection of these organs and imaging *in situ*.

The possibility to visualize biological tissue in 3D has proven to be invaluable for understanding complex processes in various tissue forms–including that of insects. For centuries, imaging at depth required the physical sectioning of tissue due to photon scattering. The imaging limit of conventional microscopy in terms of penetration depth is set by a physical parameter of photons known as the mean free path (MFP) (reviewed by [15]) which refers to the collision events of these wave-particles. With widefield epifluorescence microscopy, high quality imaging is possible when the thickness of tissue sections is within 10–50 μm (Fig 1A). With confocal and multi-photon microscopy, greater penetration depths (>500 μm) can be achieved (Fig 1B); however, this penetration depth is still impractical for highly resolved 3D digital reconstructions of large specimens.

Novel 3D imaging techniques such as optical projection tomography (OPT) [16] and light sheet fluorescence microscopy (LSFM) also known as selective plane illumination microscopy (SPIM) or ultramicroscopy, allow visualization of large objects without the need of physical sectioning [17] (see commentary by [18]). A pre-requisite for these imaging techniques applied to opaque samples is optical clearance, as in transparent media light propagates deeper into tissues, (reviewed by [15]. In order to generate a transparent sample, tissues can be chemically cleared using various solvents and imaging techniques (reviewed by [9]). After rendering the specimen transparent, OPT imaging is achieved via tissue trans- and epi-illumination over multiple projections [16] as the specimen is rotated through 360 degrees in angular steps around a single axis (Fig 1C). Virtual sections are reconstructed from the acquired images using a back-projection algorithm [19]. OPT achieves penetration depths of up to 15 millimeters [16], and allows high resolution 3D image reconstructions of the sample's complete volume. Conversely, LSFM uses a thin plane of light (or light sheet), shaped by a cylindrical lens or a laser scanner to exclusively illuminate the focal plane of the sample (Fig 1D) [17] and is

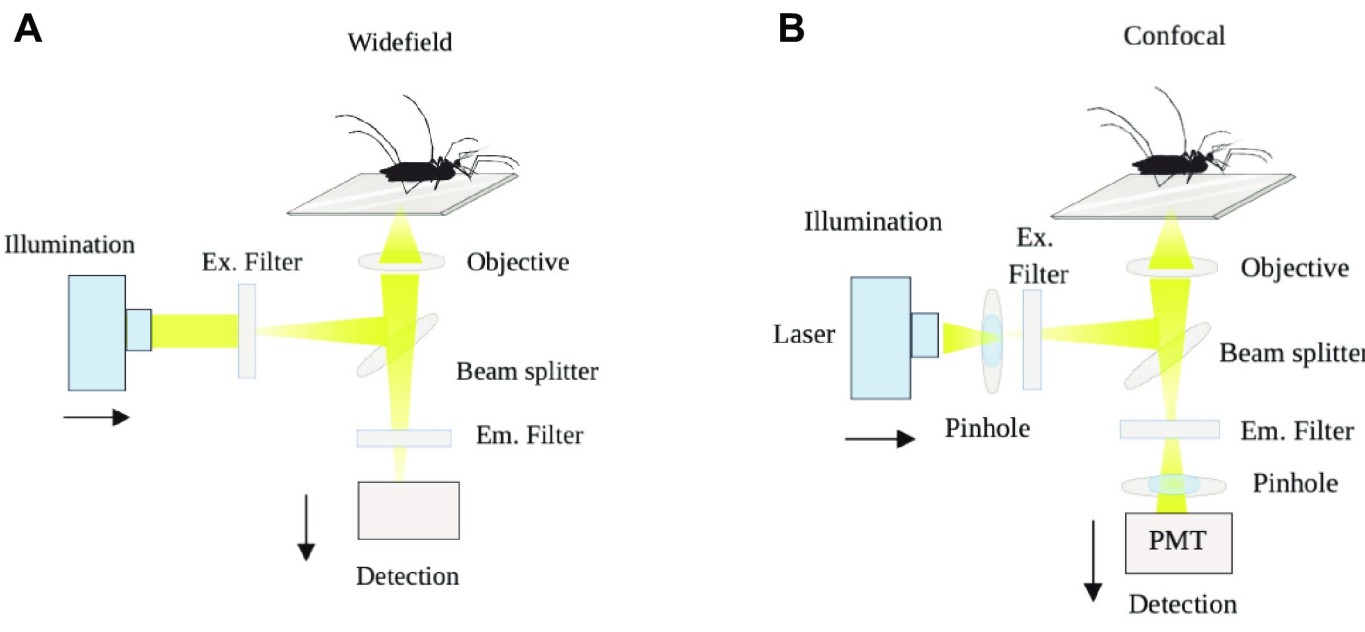

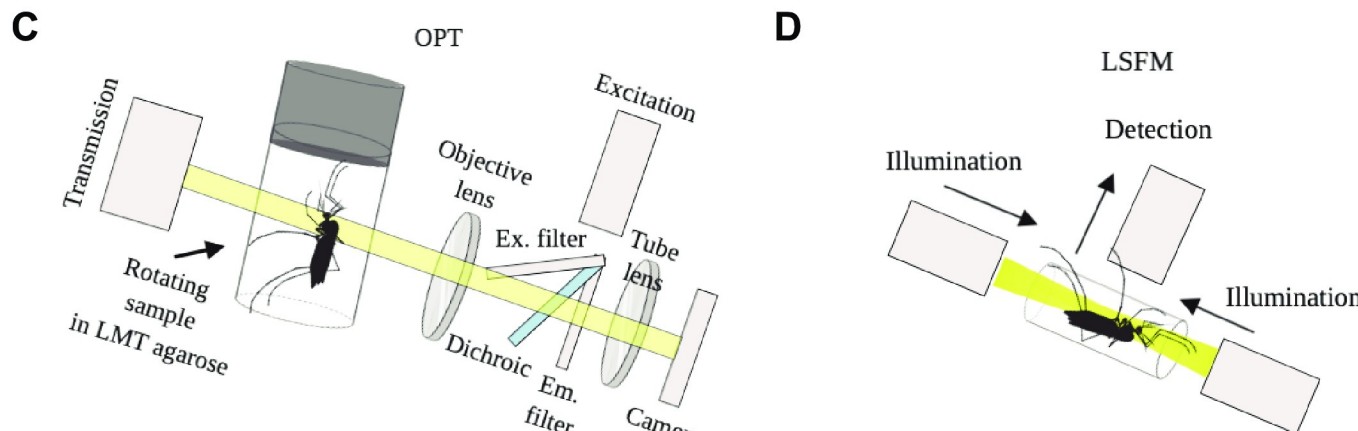

**Fig 1. Microscopy methods used for imaging mosquitoes. A)** 'Inverted' widefield microscopy: White light is filtered to the appropriate emission wavelength, and the emitted fluorescent light is projected onto a camera. **B)** Confocal microscopy: laser light is focused onto the specimen and a pinhole excludes out of focus light. Instead of a camera, photomultiplier tubes (PMTs) collect photons. **C)** Optical projection tomography: The optically cleared specimen is embedded in agarose, attached to a metallic cylinder within a rotating stage, and suspended in an index-matching liquid to reduce scattering and heterogeneities of refractive index throughout the specimen. Images are captured at distinct positions as the specimen is rotated. The axis of rotation is perpendicular to the optical axis, so that straight line projections going through the sample can be generated, and collected on the camera. **D)** Light sheet fluorescence microscopy: The sample is embedded in agarose, and suspended within a sample holder inside an index-matching liquid. A thin (μm range) slice of the sample is illuminated perpendicularly to the direction of observation. Scanning is performed using a plane of light, which allows very fast image acquisition. This figure was created using BioRender. com. Diagrams to generate this figure are republished from BioRender under a CC BY license, with permission from BioRender, original copyright (2020).

characterized by high imaging speed, reduced toxicity, and reduced photobleaching (reviewed by [20]). 3D image formation is based on raw images being assembled after translation or rotation of the entire sample. The difference between OPT and LSFM in terms of mesoscopic imaging is that OPT images are isotropic (without distortion in any 3D axis), but the focal depth is deliberately large and low numerical aperture (NA) objectives are used yielding low

resolution. Conversely, LSFM images are anisotropic (with higher resolution in the x and y axes than in *z*), but usually work with higher NA objectives and therefore achieve a high resolution, up to single cell level. OPT can also be designed for single cell resolution but at the expense of sample size imaging capacity (reviewed in [21]).

Open source, custom built-versions and free software for LSFM (OpenSPIM) [22,23] and OPT (OptiJ) [24] have been generated, making these imaging platforms easily accessible across laboratories and disciplines. OPT and/or LSFM have been used to image various specimens (reviewed in [9]) including a detailed reconstruction of the anatomy of the flight musculature of a *Drosophila* fly, its nervous and digestive systems, and ß-galactoside activity throughout the fly's whole body [25,26]. Using OPT or LSFM, fluorescence reporters and antibody labeling can be used to reveal specific structures or protein localizations. Recent work showed the development of *P. berghei* (*Plasmodium* parasites infecting mice) at fixed points in optically cleared mosquitoes using CUBIC (Clear Unobstructed Brain/Body Imaging Cocktails and Computational Analysis) [27].

Here, we generated 3D reconstructions of optically cleared *Anopheles stephensi* mosquitoes infected with mCherry- or GFP-expressing *Plasmodium berghei* parasites using OPT and LSFM. We present a comparative evaluation of different clearance protocols and discuss their value concerning different applications and research questions. Ultimately, following testing of the various protocols, we performed further work with the method we found most efficient for clearance while preserving mCherry fluorescence. Thus, the reconstructions we present are based on mosquitoes rendered transparent using Murray's clear [28,29]. Our approach provided detailed views of the anatomy of the mosquito head, thorax and abdomen. We envisage that the presented techniques will be of use for the study of pathogen and vector biology.

## Results

### Optical clearance of infected and uninfected *Anopheles stephensi* mosquitoes

A major hurdle for whole-body mosquito imaging is light scattering due to presence of the cuticle. To overcome this hurdle, we used optical clearing methods to increase light depth penetration and reduce scattering. While multiple clearance techniques have been developed over the past decade, we tested four different techniques based on either organic solvents or water, and we compared them in terms of a) time to achieve mosquito transparency (Fig 2A), b) preservation of fluorescence in full mosquitoes (Fig 2B and 2C) and excised midguts (Fig 2D) as well as c) conservation of mosquito tissue morphology (Fig 2E). These methods are BABB (Murray's clear) [28,29], Sca*l*eS [30], SeeDB [31], and 3DISCO [32]. Results are summarized in Table 1 and Fig 2. For all methods, samples were mounted as described in S1 Fig.

### Mosquito clearance and transparency was successful using 3DISCO and BABB

First, we compared tissue transparency achieved by 3DISCO, BABB, Sca*l*eS and SeeDB. Optical clearance was defined to be successful (100% transparency) as soon as imaging of the entire width of the mosquito body with OPT and confocal microscopy was possible. The two solvent-based protocols, BABB and 3DISCO, achieved clearance of the mosquito cuticle within a median time of 6.5 days (SD = 4.5; n = 50 in triplicate experiments) and 21 days (SD = 8.6; n = 50 in triplicate experiments) respectively (Fig 2A). Conversely to BABB and 3DISCO, the sorbitol-based clearance method Sca*l*eS achieved only up to 80% transparency in all mosquitoes tested, within a median time of 32.5 days (SD = 6.0, n = 50 in triplicate experiments).

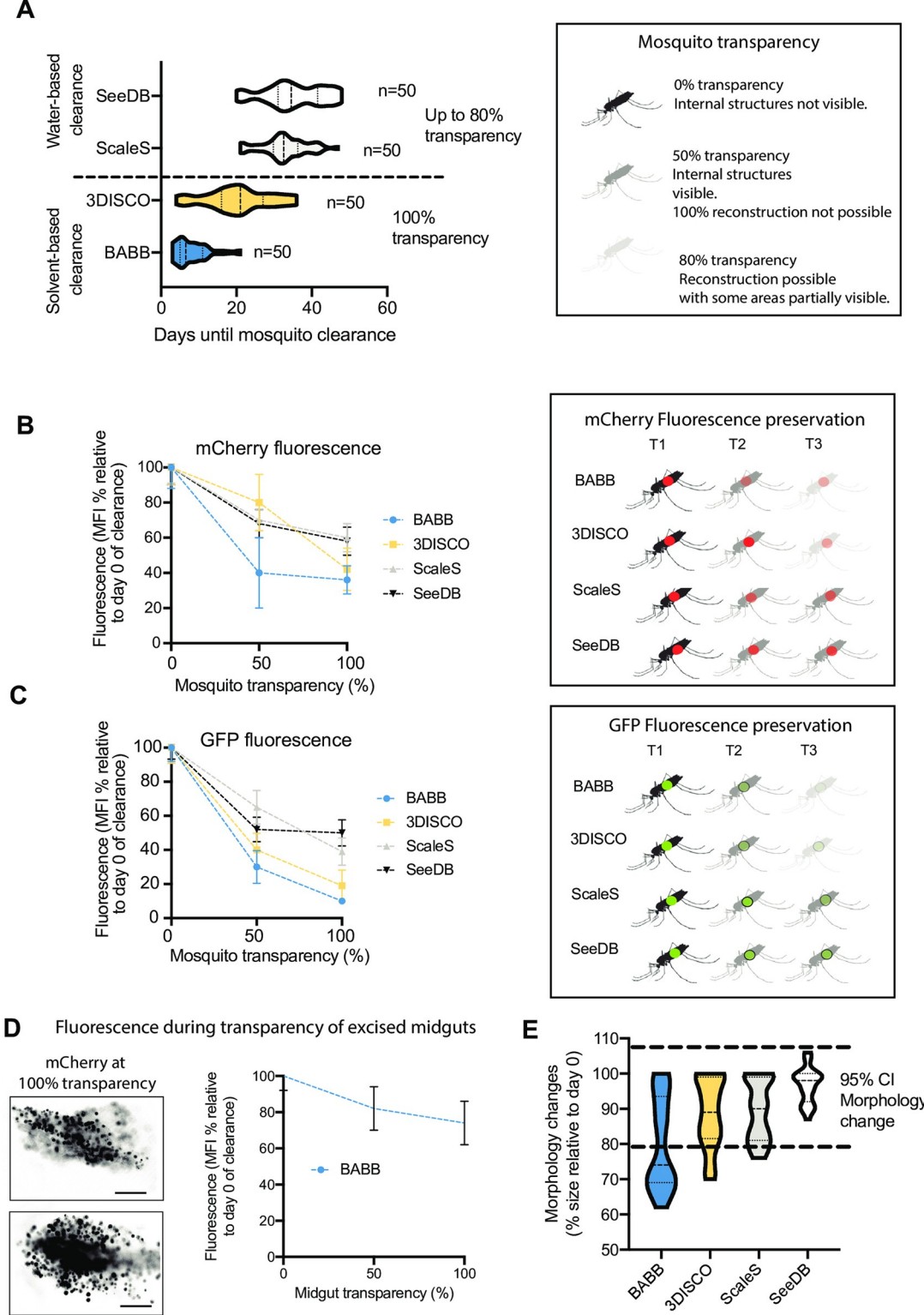

**Fig 2. Quantitative and semi-quantitative assessment of tissue clearance methods as applied to *A. stephensi* mosquitoes. A)** Determination of time of clearance for achievement of transparency of *Anopheles* mosquitoes. For each method, 50 mosquitoes were embedded in ultrapure low melting temperature agarose gel, and processed as required for SeeDB, ScaleS, 3DISCO and BABB clearance. Mosquitoes were imaged by confocal microscopy, and 100% transparency determined as the possibility to image through the full sample at a high level of detail and without significant light scattering. Two-way ANOVA test between

methods for days required for mosquito clearance, p = 0.06. **B)** mCherry fluorescence intensity and **C)** GFP fluorescence intensity were measured using a widefield microscope in all infected mosquitoes at the time of euthanasia (at day 12 post-infection) prior to clearance, and this value was defined as 100% for each sample. Fluorescence was measured again at various times of clearance. Graph B shows the average fluorescence percentage relative to time 0, at time points whereby 50% and maximum transparency were achieved. Dots represent average percentage. Error bars represent standard deviations. ANOVA tests resulted in p-values of 0.86 and 0.82 for B and C respectively. **D)** Mosquito midguts were excised and optically cleared using BABB. mCherry fluorescence intensity was then measured throughout midgut clearance. Images show fluorescence by the time the midgut was fully cleared. Results shown in the graph are the mean and standard deviations of 20 midguts measured. Scale bar: 20 μm. **E)** Semi-quantitative representation of morphological changes in mosquitoes following incubation in BABB, 3DISCO, SeeDB or Sc*ale*S. Mosquito sizes were measured at point 0 (day of euthanasia), and measured again at the time of maximum transparency. Dotted lines represent the range considered not significant, based on all measures regardless of method used. Only BABB resulted in tissue shrinkage leading to a median size decrease of 26% (SD = 13). Sample size for each method was n = 50 mosquitoes. Two-way ANOVA test between all clearance methods for morphology, p = 0.12. Mosquito diagrams in this figure are republished from BioRender under a CC BY license, with permission from BioRender, original copyright (2020).

Next, we tested SeeDB, a protocol that combines use of the water-soluble clearing agents fructose and urea. Similar to what we found for Sc*ale*S, clearance of the cuticle was only partial after 34.5 days of incubation (SD = 7.9, n = 50 in triplicate experiments) using these water-based methods (Fig 2A).

## Fluorescence preservation significantly differs among clearance methods and fluorophores used

In a next step, we compared the preservation of parasite-expressed fluorophores (mCherry or GFP) in the mosquito midgut by monitoring the emitted fluorescence until >80% clearance was reached with 3DISCO, BABB, Sc*ale*S and SeeDB (Fig 2B). Our findings for 3DISCO showed that the cuticle is fully cleared within a median time of 21 days. However, compared to untreated mosquitoes, mCherry signal was reduced by 20% (SD = 16) by the time the mosquitoes were 50% cleared, and by 58% (SD = 13) by the time mosquitoes were 100% cleared. BABB achieved fastest optical clearance, yet fluorescence decreased by 60% (SD = 20) at 50% mosquito transparency, and by 64% (SD = 12) when mosquitoes were fully transparent. Sc*ale*S and SeeDB were slowest to achieve what we defined as 100% optical clearance, yet fluorescence preservation with both methods had a tendency to be higher than with either BABB or 3DISCO. With Sc*ale*S clearance, fluorescence decreased by 30% (SD = 6.5) at 50% mosquito transparency, and by 40% (SD = 8.0) by the time mosquitoes were fully transparent. With Sc*ale*S, fluorescence decreased by 32% (SD = 8.2) at 50% mosquito transparency, and by 42% (SD = 8.5) by the time mosquitoes were fully transparent (Fig 2B).

In cleared mosquitoes harbouring GFP-expressing parasites, the loss of fluorescence was significantly higher compared to mCherry, both at 50% and 100% optical mosquito clearance. Particularly, both solvent-based methods (i.e. BABB and 3DISCO) resulted in 70–80% fluorescence loss by the time full clearance was achieved (Fig 2B and 2C). To determine specific fluorescence loss, we measured fluorescence intensity throughout clearance time in excised mosquito midguts. The time for achieving transparency in midguts was half of that needed to

**Table 1. Comparison of clearing methods for mosquito cuticle.**

| Method | Principle | Time to achieve tissue transparency | Preservation of fluorescence signal | Preservation of tissue morphology |
|---|---|---|---|---|
| **3DISCO** | Organic solvent | 20–30 days (++) | + | Unaltered |
| **BABB (Murray's clear)** | Organic solvent | 5–10 days (+++) | + | Slight dehydration |
| **ScaleS** | Water-based | Partial clearance at 30 days (+/-) | +++ | Unaltered |
| **SeeDB** | Water-based | Partial clearance at 30 days (+/-) | +++ | Unaltered |

achieve transparency of full mosquitos, and fluorescence intensity was better preserved, as shown in Fig 2D. Data shown are the result of measuring 20 midguts at day 8–10 post-feed.

## Different clearance methods conserve mosquito morphology equally well

Clearance methods can introduce morphology artefacts, including dehydration or expansion of biological samples. To determine the morphological alterations introduced by each of the methods tested, we measured relative size change of the samples by the time of maximum optical clearance. We found that 3DISCO, SeeDB Sca*le*S and BABB induced slight morphological changes in the samples, with the median size being 89% (SD = 10), 90% (SD = 8.5), 98% (SD = 5.0), and 77% (SD = 13) the size of the same samples prior to clearance, respectively (range of significance is shown between dotted lines, Fig 2E).

Considering all parameters, we chose the method with the greatest tissue clearance success within a short time-frame, and therefore we decided to use BABB as the method of choice for all subsequent experiments reported in this work. Our rationale for this choice is that in subsequent work we will aim at method optimization for fluorescence preservation, however we considered that tissue transparency was a significant advantage for various relevant anatomical observations without the need of fluorophores, and BABB was the most efficient method to achieve this.

## OPT enables visualization of the entire anatomy of intact adult mosquitoes

Cleared adult *Anopheles stephensi* mosquitoes were three-dimensionally reconstructed from OPT projections (Fig 3A and 3B; S1 File), to represent various features of the head, the thorax, and the lower body including the midgut *in situ*. Following clearance, the absorption, reflection and auto-fluorescence of the cuticle were reduced to an extent that internal organs of the mosquito could be visualized (Fig 3). The mosquito head is specialized for processing sensory information, and feeding. The mosquito has compound eyes made up of multiple lenses called ommatidia, which could be faithfully reconstructed by OPT (Fig 3C, top panels). Moreover, olfaction is an additional primary sensory modality of mosquitoes. OPT enabled imaging of the antennae, and the mouthparts (Fig 3C, top panels). (marked with arrowheads). The *Anopheles* mosquito thorax is specialized for locomotion, and is divided into three segments, the prothorax, the mesothorax, and the metathorax, all of which were readily distinguished by OPT (Fig 3B; note that the surface rendered image has been previously shown in [9]). Each thoracic segment supports a pair of legs (3 pairs in total), while the mesothorax additionally bears a pair of wings (Fig 3C, bottom panels) (Fig 3A and 3B). Finally, the abdomen (Fig 3A and 3B) is specialized for food digestion, reproduction, and egg development. The *Anopheles* mosquito abdomen is long and can be divided into up to 10 segments, clearly visible by OPT (Fig 3B, right panel). Unlike the thorax, segments I to VIII can expand significantly upon ingestion of a blood meal. This expansion was clearly visible in fed mosquitoes. Segment VIII bears the terminal anus of male and female mosquitoes. In females, segments IX and X bear the gonopore, and a post-genital plate, while in males, segments IX and X harbour a pair of clawed claspers and an aedigus. All structures of the thorax, abdomen and reproductive segments were readily visualized by OPT (Fig 3A and 3B) and are marked by arrows respectively.

## OPT enables imaging *Plasmodium* parasites within the isolated midguts and salivary glands

We used mCherry- or GFP-tagged *P. berghei* to observe parasite distribution within entire mosquitoes at various times post blood-feed, however, encountered significant

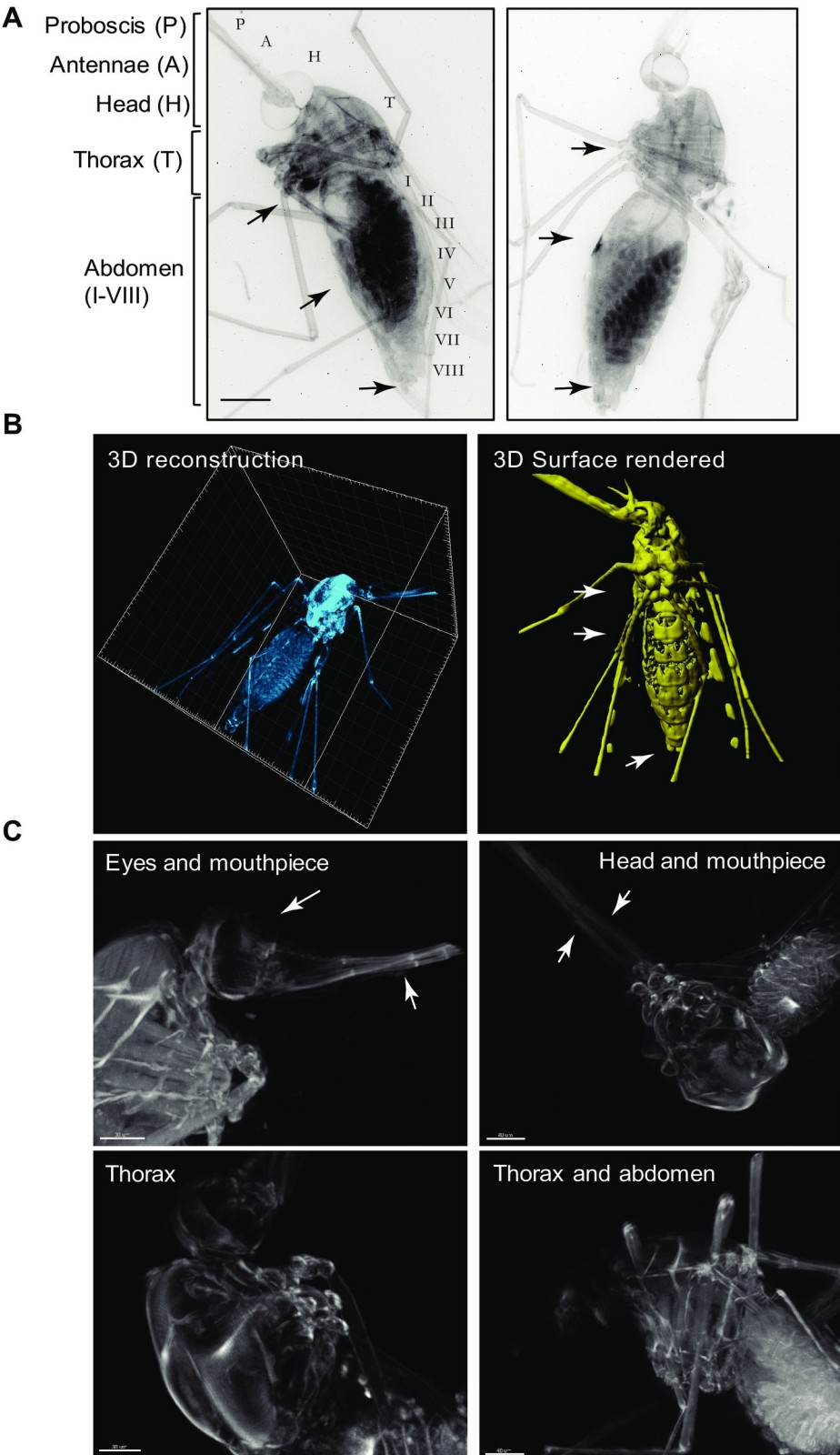

**Fig 3. Visualization of an optically cleared *Anopheles stephensi* female mosquito. A)** 2D reconstruction of two reconstructed mosquitoes showing detailed views of the head structures (H) including the antennae (A), and the proboscis (P) and the eyes. Detailed view of the thorax is also possible, as well as of all segments of the abdomen. Scale

bar: 500 μm. **B)** 3D reconstruction and clear view of all body cavities of an optically cleared mosquito (left panel). 3D reconstruction and rendering of the mosquito (right panel) clearly showing abdominal segments, thorax and head features (previously shown in [9]); see S1 File. **C)** Close-up views of various views of the optically cleared mosquito body including the eyes and mouthpieces (side view, upper left panel), the head and mouthpiece (top view, upper right panel), the thorax (side view, lower left panel), and the abdomen including eggs (side view, lower right panel). Scale bar: 200 μm. All elements of this figure were generated by the authors.

autofluorescence arising from the eggs. We imaged isolated mosquito midguts by LSFM (Fig 4) and performed an OPT time course experiment using intact mosquitoes.

Immediately after a blood-feed on an infected mouse, no fluorescent signal from parasites was detected using OPT, yet the mosquito anatomy could be visualized at high level of detail. At day 16 post-infection, we detected strong mCherry signals in the salivary glands, the meso-thorax, the base of the wings, and the midgut (Fig 4A). However, the signal was diffuse and did not allow for detection of individual sporozoites or oocysts in the complete mosquito. Also detailed insights into multiple mosquitoes imaged shows strong signal arising from autofluor-escence which in some cases is indistinguishable from mCherry-specific fluorescence and in some cases is not (S2 and S3 Figs). In contrast, LSFM performed on isolated midguts clearly shows individual *P. berghei* oocysts across a full rotation of the sample (Fig 4B and S2 File). In S2 Fig we show that detection of specific fluorescence as opposed to autofluorescence is

**Fig 4. Visualization of *Plasmodium*-infected *Anopheles stephensi* female mosquitoes. (A)** 3D project (B&W) and 3D reconstructions of mosquitoes at the beginning and end of *P. berghei* infection as well as egg development (yellow rendering). Scale bar 500 μm. **(B)** Isolated *P. berghei*-infected mosquito midguts imaged by LSFM. Oocysts are shown in white (*P. berghei*-mCherry) or black (*P. berghei*-GFP). Scale bars: 100 μm. All elements of this figure were generated by the authors.

possible using OPT, and we suggest that LSFM or hybrid methods such as OPTiSPIM could be well suited for specific quantification of oocyst numbers and sizes in whole body mosquitoes. In excised cleared midguts, the technique we present here, allowed full quantification of oocyst numbers. Moreover, in optically cleared mosquitoes, we found it would also be possible to perform egg quantification in the entire mosquito.

## Discussion

One of the major hurdles for whole-body imaging of insects is light scattering due to presence of the cuticle. Optical clearing techniques enable an increase of light depth penetration and generally reduce light scattering by replacing cellular water with solutions that have a refractive index similar to that of the cell membrane. Lipids in cell membranes are dominant scattering agents in biological tissues, and optical clearing methods can obtain approximately uniform refractive index profiles by removing them. Reduced light scattering ultimately leads to higher spatial resolution and greater contrast. Various clearance techniques have been developed, including the use of organic solvents [29,32–34], water [30,31,35], and electrophoresis-based protocols [36,37]. In this study, four techniques were tested for mosquito clearance, and were compared in terms of a) time to achieve tissue transparency, b) preservation of fluorescence signal in mosquitoes infected with mCherry-expressing *Plasmodium berghei* parasites (albeit not effective preservation of GFP signal) and c) resulting mosquito tissue morphology following treatment. Altogether, we conclude that the use of BABB was most useful for our purposes for achieving the highest transparency in the shortest amount of time. This method allowed full clearance of the mosquito and visualizing overall anatomical features, however, it did not reach sufficient detail to distinguish individual parasites due to loss of fluorescence. In dissected midguts, however, clearance allowed high resolution imaging of oocysts. We hence present the method as a baseline for future optimization and visualization of antibody-labeled or fluorescently-tagged anatomical structures of the mosquito, relevant in the context of *Plasmodium* infections and for other insects as a starting point. Since the development of our work, alternative methods, including CUBIC [34,38], have been developed and used in various contexts with successful fluorescence preservation, which might also prove useful for studying host-pathogen interactions using fluorescent probes.

Our aim while developing the clearance and full body/ full midgut imaging methods by LSFM and OPT was to provide a complementary tool to currently existing methods, that addresses some shortcomings that current methods for mosquito imaging pose. These methods are summarized in Table 2 (further discussed in [9]), and can be separated into two groups based on their advantages and disadvantages: the first group is made of techniques that have allowed imaging parasites within mosquitoes with great detail and high resolution (including TEM, immunohistochemistry, and confocal microscopy). The main shortcoming of these methods, however, is that for visualization of the entire mosquito body, extensive and laborious physical sectioning is required, as well as laborious 3D reconstructions with possible artifacts deriving from the physical sectioning itself. Tissue optical clearance methods, as well as OPT and LSFM have been proposed in various contexts as an optimal tool to gain insight into a full organism without the need for physical sectioning and complex stereology techniques. The second group of tools used for mosquito imaging includes two-photon microscopy, widefield microscopy, scanning electron microscopy (SEM) and synchrotron X-ray tomography. These tools have provided important findings on the mosquito anatomy, and parasite development. While these tools have allowed full body imaging, two-photon and widefield imaging still have limited tissue penetration due to the mosquitos' opaque cuticle. Conversely, SEM requires relatively complex sample preparation compared to the relatively simple one required

**Table 2. Comparison of OPT/LSFM with other microscopy procedures.**

| Method | Principle | Advantage | Disadvantage |
|---|---|---|---|
| **Widefield microscopy** | Light passes through the sample, maximizing illumination | Simple to perform. Fluorescence detection possible. Live imaging possible. | Does not allow acquisition of detailed parasite development or localization. For this, it would require physical sectioning. |
| **Confocal microscopy** | Increases optical resolution by means of a pinhole that blocks out of focus light. | Higher optical resolution possible. Visualization of specific structures and their interactions possible. Live imaging possible. | Requires optical sectioning. 3D reconstruction of a full sample is time consuming. If sample uncleared, scattering is problematic. |
| **Two photon microscopy** | Two low energy photons cooperate to cause a higher-energy electronic transition in a fluorescent molecule. | Penetration of up to 1mm of depth, and minimization of phototoxicity. Live imaging possible. | Requires optical sectioning. 3D reconstruction of a full sample is time consuming. If sample uncleared, scattering is problematic. |
| **Electron microscopy** | Uses beam of accelerated electrons as a source of illumination. | Very high resolution achievable. Information on details of structures, tissues, cells, organelles and sub-organellar structures easy to obtain. | Sample cannot be live. Method for sample preparation is complex and time consuming. Full mosquito reconstruction would be very time consuming. |
| **OPT** | Form of tomography involving optical microscopy that allows full 3D sample reconstruction. | Fluorescence based method. Allows detailed visualization and 3D reconstruction. Does not require physical sectioning of the sample. | Requires optimization of tissue clearance and fluorescence preservation. At the moment cannot be used in live samples. |
| **LSFM** | Sample scanning with a plane of light. | Fluorescence based method. High optical resolution and high acquisition speed. Allows detailed visualization and 3D reconstruction. Does not require physical sectioning of the sample. | Requires optimization of tissue clearance and fluorescence preservation. At the moment cannot be used in live samples. |
| **Synchrotron X-ray tomography** | Based on the detection of the attenuation or phase shift of the beam transmitted through a sample. | Allows visualizing the interior of bodies in a non-destructive manner. High SNR at short time scales. | Complex setup. Requires specialized training. |

for OPT or LSFM imaging. Equally, synchrotron X-ray tomography requires a complex platform for imaging, while OPT and LSFM can be performed with relatively simple setups and require relatively little user training. Moreover, optical transparency is compatible with fluorescence and autofluorescence, which increases the range of structures/cells of interest that can be simultaneously visualized to answer specific questions of interest. For example, we envisage that imaging of intact optically cleared mosquitoes using the absorption mode will also enable tracking of pathogen-induced changes in gene expression [16], and changes in expression of specific components in the mosquito's sensory systems.

While the work we present here provides a baseline for optical clearance, and demonstrates in practice the principle using different fluorophores, we expect that groups interested in using this tool can envisage investigating a wide range of physiologically relevant questions. For instance, we know that changes in insect sensory responses and behavior are likely to increase the chances of parasite transmission, and are thought to arise either from changes in the expression of salivary gland components [39–41], or from the modulation of the mosquito nervous system. These changes may be induced by parasites, including *Plasmodium* [42]. Imaging of specific molecules and gene expression levels at a whole body level that potentially influences mosquito behavior but could be achievable with OPT or LSFM. Our work was successful in the clearance of the mosquito thorax and visualization of internal structures. Further study of these structures in undissected mosquitoes might shed light into the mosquito's biology and vectorial capacity. Beyond specific optimization of OPT and LSFM for answering specific physiological questions of interest, we also envisage that the use of combined tools such as the hybrid system OPTiSPIM [43], could be relevant to study parasites *in vivo*. Altogether, another advantage we see in OPT/LSFM is the possibility of monitoring the entire body of the mosquito during infection (rather than specific sites at a time). A limitation we faced in our study was the overlap between autofluorescence arising from either the blood-meal or the

eggs, and parasite-specific signals. While in some cases we were able to separate them, further optimization is required to completely achieve separation. We recommend for instance, the use of proteins with fluorescence in wavelengths of far red (e.g. 647 nm) rather than those closer to the autofluorescence signal (in the range of 450–550 nm). Another way to solve this shortcoming is through better preservation of GFP or mCherry fluorescence which would then allow differential thresholding of signals for quantification. Also, autofluorescence can be somewhat limited by using wider-pass filters.

Although some tissue clearance methods introduce effects including swelling or shrinkage, from the images we were able to obtain in dissected midguts, we observed no significant changes in parasite morphology. However, we encourage that for each application, relevant controls (e.g. parallel imaging methods) should be acquired to ensure no artifacts are introduced.

In conclusion, we have shown that adult *Anopheles* mosquitoes can be cleared efficiently, and that this allows for transmission of white and fluorescent light to detect anatomical features of parasite-infected mosquitoes in 3D using OPT and LSFM. In the future, another aim based on tissue clearance that we think will be extremely useful in the context of vector imaging is the possibility of imaging parasite development in living mosquitoes. Finally, altogether, while several years ago the amount of data obtained from 3D reconstructions would be significantly large and difficult to analyze, image analysis has advanced at increased speed over the past few years. Various tools now exist [44–46] that facilitate the analysis of large datasets such as the ones obtained in our work. We hope that the data presented here will hopefully inspire the development of whole-body imaging technologies to allow for the discovery of important host-pathogen interactions in the malaria field.

## Materials and methods

### Ethics statement

Mouse infections were carried out under the approval of the Animal Research Ethics Committee of the Canton Bern, Switzerland (Permit Number: 91/11 and 81/11); the University of Bern Animal Care and Use Committee, Switzerland; and the German Tierschutzgesetz (Animal Rights Laws). We have followed the Ethical Guidelines for the Use of Animals in Research. For all mosquito feeds, female mice 5–8 weeks of age, weighing 20–30 g at the time of infection were used. Mice were purchased from Harlan or Charles River laboratories. Blood feeding to mosquitoes was performed under ketavet/dorbene anaesthesia, and all efforts were made to minimize animal suffering.

### Parasites lines and their maintenance in mosquitoes

*P. berghei*-ANKA lines were used in this study to infect mice used for mosquito feeds. *P. berghei*-mCherry$_{Hsp70}$ [47], *P. berghei*-GFP$_{Hsp70}$ and PbmCherry$_{Hsp70}$FLuc$_{ef1\alpha}$ [48] express fluorescent mCherry that localizes to the cytosol of the parasite, and is expressed constitutively throughout the parasites' life cycle.

Balb/c mice were treated with phenylhydrazine two days prior to intra-peritoneal (i.p.) infection with *P. berghei*-mCherry$_{Hsp70}$ or PbmCherry$_{Hsp70}$FLuc$_{ef1\alpha}$. After 3 days of infection, gametocyte exflagellation was assessed. Upon confirming exflagellation, the infected mice were used to feed various cages with 100–150 *Anopheles stephensi* female mosquitoes. Mice were anaesthetized with a combination of Ketasol/Dorbene anaesthesia, and euthanized with $CO_2$ after completion of the feed. Afterwards, mosquitoes were fed until use, with 8% fructose containing 0.2% PABA.

## Mosquito embedding

Adult female *Anopheles stephensi* mosquitoes were killed at various times following feeds on mice infected with *P. berghei*, and fixed overnight at 4°C in a 1:1 mixture of 4% paraformaldehyde in 1x PBS and 100% ethanol. Mosquitoes were then washed 3 times in 1xPBS for 5 minutes each time. Washed mosquitoes were embedded in 1.3% ultrapure low-melting agarose (Invitrogen) in deionized water. Gels containing the mosquitoes were transferred for at least 2h to 4°C. Using a single-edge blade, the gel was then trimmed into a block containing a single mosquito in the centre.

## BABB (Murray's clear)-based mosquito dehydration and clearance

Agarose blocks containing the mosquitoes at all times post blood-feed (including 2 h, 20 h, and day 1 through day 16), were dehydrated in a graded ethanol series (50, 70, 90, 96, and 100%) for 1 h each. Mosquitoes were then transferred to another flask containing 100% ethanol, and dehydrated overnight. Finally, mosquitoes were incubated in a clearing solution consisting of two parts benzyl benzoate and one part benzyl alcohol (BABB, also known as Murray's clear) [29,49,50] for at least 10 days, until they became transparent.

## 3DISCO-based mosquito dehydration and clearance

Previous work showed that tetrahydrofluoran (THF) in combination with dibenzyl ether (DBE), fully clears multiple mouse tissues including the lymph nodes, spinal cord, lungs, spleen and brain, while successfully preserving fluorescent signals [32,33]. Two clearing protocols were adapted for use in mosquitoes, namely a relatively short protocol consisting of dehydration in a graded THF series (50, 70, 80 and 100%) for 30 minutes each, followed by 2 further 30 minute incubations in 100% THF. This was followed by a 20-minute incubation in dichloromethane (DCM), and a 15-minute incubation in DBE. The long protocol consisted on dehydration in the graded THF series for 12 h each, followed by 2x 12 h incubations in 100% THF. This was followed by clearance in DBE, without the intermediate DCM step.

## SeeDB-based mosquito dehydration and clearance

In 2013, Ke and colleagues [31] first published a water-based optical clearing agent called SeeDB, which had the advantage of preserving fluorescence, including that of lipophilic tracers, while also preserving sample volume and cellular morphology. In order to prepare fructose solutions, D(-)-fructose was dissolved in distilled $H_2O$ at 65°C, and upon cooling to 25°C, α-thioglycerol was added to give a final concentration of 0.5%. Mosquitoes were initially fixed in 4% PFA, and embedded into 1% ultrapure agarose in $dH_2O$. Fixed mosquitoes were then serially incubated in 20%, 40% and 60% fructose, each for 4 h, followed by a 12 h incubation in 80% fructose, a 12 h incubation in 100% fructose, and incubation in SeeDB at either 37°C or 50°C.

## Microscopy—Optical projection tomography (OPT)

OPT scanning was performed according to the manufacturer's instructions (Bioptonics). Filter sets were exciter 425/40, emitter LP475 for autofluorescent signal, exciter 480/20, emitter LP515 for green fluorescent signal; and exciter 545/30, emitter 617/75 for red fluorescent signal. Raw data were converted into 3D voxel datasets using NRecon software from Bioptonics. Reconstructed virtual *xyz* data sets were exported as.tif files and analyzed with IMARIS (Bitplane) for visualization and/or isosurface reconstruction of parasite distribution in the

mosquitoes. IMARIS reconstructions were carefully adjusted to fit original NRecon reconstructions.

## Light sheet fluorescence microscopy (LSFM)

Light sheet fluorescence microscopy scanning was performed using a commercially available Ultramicroscope system (LaVision BioTec). Light was produced by a 200-mW laser that illuminates the sample from both sides by two co-localized thin sheets of light to compensate for absorption gradients within the tissue. A 10x objective with a NA of 0.3 was used.

## Microscopy—Confocal imaging

Confocal imaging of dissected midguts and salivary glands for validation of the observations performed by OPT and LSFM was performed using a Leica SP8-STED microscope. Midguts and salivary glands were imaged using a 20x air objective, using a white light laser at a wavelength of 550nm, and a 63x oil immersion objective using a white laser at wavelengths 405 and 488 nm. The LASX software was used for image acquisition.

## Sample mounting for OPT and LSFM

Microscope setups for conventional widefield and confocal systems are remarkably different to those of OPT and LSFM (Fig 1). For conventional fluorescence microscopy samples are usually placed on glass bottom dishes or microscope slides in which they are overlaid with a coverslip. Preparation for OPT and LSFM requires placing the sample in a medium- or liquid-filled chamber that enables rotation or motion during image acquisition (S1A Fig). In order to take full advantage of the 3D imaging technique all the specimens need to be mounted into a special metal sample holder that is inserted into the chamber from a magnet above (S1B Fig). The specimen may be embedded in a gel such as low melting agarose dissolved in the medium or buffer of choice (S1C Fig). The medium keeps the sample in place without influencing the penetration of light and imaging quality.

## Supporting information

**S1 Fig. Mosquito mounting and embedding. A)** OPT imaging requires embedding the mosquito in low-melting temperature ultrapure agarose gel, and mounting it onto a metallic cylinder that is attached to a rotating stage via a magnet. The embedded attached mosquito is then lowered into a chamber containing index-matching liquid, such as Murray's clear medium. The setup for Ultramicroscopy imaging involves embedding the mosquito in low-melting temperature ultrapure agarose gel, and mounting it on a lower ring of the customized holder. Both the holder and the embedded mosquito are submerged into a chamber containing index-matching liquid. **B)** Methods for mounting mosquitoes to enable imaging and rotation. **C)** Petri dishes showing (1) fixed mosquitoes prior to optical clearance and embedding and (2) optically cleared mosquitoes embedded in ultrapure low-melting temperature agarose. S1A Fig was created using BioRender.com.
(PDF)

**S2 Fig. Detected fluorescence and autofluorescence signals in undissected mosquitoes.** Given the very successful clearance obtained with BABB, fluorescence quenching occurs. We show in this panel various possible outcomes of clearance using BABB, including **A)** a mixture of detectable fluorescence in the midgut (yellow arrows), clear autofluorescence arising lower in the body (green arrows) and autofluorescence arising from eggs (blue arrows); **B)** clear autofluorescence arising from the eggs, but no other detectable signal in the abdomen; **C)**

indistinguishable abdominal signal, without the possibility of distinguishing the bloodmeal from the eggs and potential parasites in the midgut. All figures were generated by the authors of this manuscript.
(PDF)

**S3 Fig. Specific fluorescence.** Examples obtained from S2 Fig, showing separate autofluorescence and mCherry signal, demonstrating preservation of mCherry. All figures were generated by the authors of this manuscript.
(PDF)

**S1 File. 3D visualization of an optically cleared *Anopheles stephensi* female mosquito, imaged by optical projection tomography.** This video was generated by the authors of this manuscript.
(AVI)

**S2 File. 3D visualization of an optically cleared *Anopheles stephensi* mosquito midgut, imaged by LSFM.** Fluorescent bodies correspond to *Plasmodium* oocysts. This video was generated by the authors of this manuscript.
(AVI)

## Acknowledgments

We are grateful to Volker Heussler (IZB, University of Bern) for his intellectual input and funding of this project. We thank Uroš Kržič for acquisition and processing of Movie 2 included in this manuscript. We thank Renzo Danuser (Theodor Kocher Institute, University of Bern) for important input for the OPT technique, and Leandro Lemgruber (Wellcome Trust Centre for Molecular Parasitology) for his input on image processing. OPT and confocal microscopy was performed on equipment supported by the Microscopy Imaging Center, University of Bern, Switzerland. We thank Ernst Stelzer and Pavel Tomancak for motivation to use SPIM.

## Author Contributions

**Conceptualization:** Mariana De Niz, Jessica Kehrer, Emmanuel G. Reynaud.

**Data curation:** Mariana De Niz, Jessica Kehrer.

**Formal analysis:** Mariana De Niz, Emmanuel G. Reynaud.

**Funding acquisition:** Mariana De Niz, Friedrich Frischknecht.

**Investigation:** Jessica Kehrer, Federica Moalli, Emmanuel G. Reynaud.

**Methodology:** Mariana De Niz, Jessica Kehrer, Federica Moalli, Emmanuel G. Reynaud, Jens V. Stein, Friedrich Frischknecht.

**Resources:** Nicolas M. B. Brancucci, Emmanuel G. Reynaud, Jens V. Stein, Friedrich Frischknecht.

**Software:** Jens V. Stein.

**Supervision:** Emmanuel G. Reynaud, Jens V. Stein, Friedrich Frischknecht.

**Validation:** Mariana De Niz, Jessica Kehrer, Federica Moalli.

**Visualization:** Mariana De Niz, Nicolas M. B. Brancucci, Federica Moalli.

**Writing – original draft:** Mariana De Niz, Friedrich Frischknecht.

**Writing – review & editing:** Mariana De Niz, Jessica Kehrer, Nicolas M. B. Brancucci, Federica Moalli, Emmanuel G. Reynaud, Jens V. Stein.

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
