## [Decision Letter · Decision Letter 0]

2 Aug 2020

PONE-D-20-22239

3D imaging of undissected optically cleared Anopheles stephensi mosquitoes and midguts infected with Plasmodium parasites

PLOS ONE

Dear Dr. De Niz,

Thank you for submitting your manuscript to PLOS ONE. After careful consideration, we feel that it has merit but does not fully meet PLOS ONE’s publication criteria as it currently stands. Therefore, we invite you to submit a revised version of the manuscript that addresses the points raised during the review process.

Overall the reviewers were positive about your study. However, both reviewers indicated that some of the figure legends need additional detail/modification, and both felt some additions to the discussion would make this a stronger paper. Please address these and their other minor concerns and return a revised copy of the manuscript as outlined below.

We look forward to receiving your revised manuscript.

Kind regards,

Photini Sinnis, M.D.

Academic Editor

PLOS ONE

Journal Requirements:

2. We note you have included a table to which you do not refer in the text of your manuscript. Please ensure that you refer to Table 2 in your text; if accepted, production will need this reference to link the reader to the Table.

Reviewers' comments:

Reviewer's Responses to Questions

**Comments to the Author**

1. Is the manuscript technically sound, and do the data support the conclusions?

Reviewer #1: Yes

Reviewer #2: Partly

2. Has the statistical analysis been performed appropriately and rigorously? 

Reviewer #1: Yes

Reviewer #2: Yes

3. Have the authors made all data underlying the findings in their manuscript fully available?

Reviewer #1: Yes

Reviewer #2: Yes

4. Is the manuscript presented in an intelligible fashion and written in standard English?

Reviewer #1: Yes

Reviewer #2: Yes

5. Review Comments to the Author

Reviewer #1: Some of the major problems of preparing mosquitos for microscopy are related to the cuticle’s biological properties, including the thickness, light scattering, extent of cuticle sclerotization, and autofluorescence. These properties make difficult the acquisition of live or fixed images of insects in general, especially those vectors of diseases. Recently, these problems have been overcome by adapting chemical clearing protocols for to increase the transparency of Anopheles mosquito cuticle, allowing acceptable preservation of both mosquito and Plasmodium parasites. For morphological studies to bring a novel conceptual advance, few characteristics must be attended, including good preservation of parasites and mosquito tissues, and the capacity to study host-parasite interactions in whole organisms. In this manuscript, De Niz et al., address some of these limitations by combining the use of chemical clearance methods with the power of Optical projection tomography (OPT) and light sheet fluorescence microscopy (LSFM) to successfully 1) obtain optically cleared Anopheles mosquitoes, 2) obtain 3D reconstruction of whole mosquitoes and 3) image Plasmodium parasites on infected midguts. Overall, the paper is well-executed, methodologically sound and well-written. Some figures can benefit from additional labeling to better guide the reader on the points emphasized by the authors (see comments). If further developed, this technology has the potential of establishing new protocols for studying host-parasite interactions.

Major comments:

Most of the discussion is a repeat of the results. The discussion would benefit from a better comparison, based on their results, of the benefits and advancement that this technology provides over other methods previously used to image infected and non-infected Anopheles mosquitoes.

A key question that the authors need to address in the discussion is what new information one can learn, given the amount of work that takes to process the samples through chemically clearance? It is not clear that the resolution level achieved allows a detailed study of host-parasite interactions. The distribution of oocysts in the midgut can be easily studied ex-vivo with well-established protocol that are less laborious, time-consuming and produce similar results. These should be discussed.

One point of concern is whether this protocol introduces morphological changes in the parasite? The low resolution of the images didn’t allow a proper evaluation of the issue.

Minor comments:

Line 170: “ScaleS and SeeDB were slowest to achieve optical clearance, yet fluorescence preservation with both methods was significantly higher than with either BABB or 3DISCO.”

The results describe a significant difference however neither the Figure 2B nor the statistics in the figure legend show statistical significance.

Line 209: “OPT enabled imaging of the antennae, the mandibles and maxillae lining the alimentary canal, and the maxillary palps (Figure 3C, top panels). OPT also allowed detailed visualization of the proboscis and its structures (all structures mentioned above are marked with arrowheads). ”

The mandibles, macillae, and maxillary palps are not clearly distinguished in these images. The arrows just point to the mouth parts but there is not description in the legend about the parts referenced in the results text.

Line 216: “Moreover, the thorax harbors the dorsal blood vessel, the tracheal and dorsal tubes (or heart), the foregut, and various nerve ducts (Figure 3A-3B).”

These structures are not clear in the figure. It will be helpful if the authors include additional arrows or marks pointing at each structure referenced in the results. In addition, it is not clear to what structures the three black arrows are point to in Fig. 3A.

Line 241: “In excised cleared midguts, our technique allowed for the first time, full quantification of oocyst numbers throughout parasite development.”

The claim that their technology allows for the first-time full quantifications of oocysts is not accurate. Full quantification of oocyst numbers in dissected midguts has been done by several methods.

Movie 2: Slow down the speed of rotation to allow for better appreciation of the image.

Reviewer #2: De Niz et al nicely described different microscope techniques and compared tissue clearing methods on Anopheles mosquito. The method comparison looked at time took to clear the mosquitoes, fluorescence preservation and morphological changes which are the key factors to look at in tissue clearing. The summary tables help to understand the comparisons. The 3D-reconstruction and surface rendering from OPT imaging were a beautiful work.

There are two points that I would like to point out.

Figure 2

It is not clear which microscope was used to quantify fluorescence preservation in figure legend and result section.

Did you look into detail if you could see individual oocysts in any tested methods and assess the level of autofluorescence?

Figure 4

It is not clear what yellow rendering indicates in figure legend (line 715 to 716). It seems surface rendering of mCherry signaling. From figure legend, it could be both mcherry and egg development.

Figure 4A, S2 & S3 show autofluorescence issue and it is hard to see individual oocysts in mosquito and identifying salivary gland sporozoite signals. For OPT and LSFM imaging, it is important to have distinguishable signals to identify the localization of protein and gene of interests. It would be great to discuss possibility to solve this issue in discussion. Not only further developing tissue clearing techniques but also the wavelength using for imaging, possibility to combine with immune-fluorescence staining advantage and disadvantage etc.

Further applications (looking at sensory response, other pathogens in abdomen) using OPT and LSFM are well described in discussion and it is a great method to use.

One minor comments

line 241-244:

Have you quantified number of eggs in mosquito or dissected oocysts? (If so, how did you count it?)

If this meant for possible application, it would be better in discussion section and discuss along with possible analysis strategies.

---

## [Author Response · Author response to Decision Letter 0]

4 Aug 2020

We have uploaded a point-to-point letter addressing the reviewer's comments.

---

## [Editor Report · Decision Letter 1]

11 Aug 2020

Dear Mariana et al.,

Thank-you for your thorough response to the reviewers. This is a nice study that will be very helpful to those working on mosquitoes and likely other vectors.

All the best,

Photini

3D imaging of undissected optically cleared Anopheles stephensi mosquitoes and midguts infected with Plasmodium parasites

PONE-D-20-22239R1

Dear Dr. De Niz,

We’re pleased to inform you that your manuscript has been judged scientifically suitable for publication and will be formally accepted for publication once it meets all outstanding technical requirements.

Kind regards,

Photini Sinnis, M.D.

Academic Editor

PLOS ONE
---

## [Editor Report · Acceptance letter]

24 Aug 2020

PONE-D-20-22239R1 

**3D imaging of undissected optically cleared**
*Anopheles stephensi*
**mosquitoes and midguts infected with**
*Plasmodium*
**parasites**

Dear Dr. De Niz:

I'm pleased to inform you that your manuscript has been deemed suitable for publication in PLOS ONE. Congratulations! Your manuscript is now with our production department. 

Kind regards, 

on behalf of

Dr. Photini Sinnis 

Academic Editor

PLOS ONE